

# Time-stamp correction of magnetic observatory data acquired during unavailability of time-synchronization services

Pierdavide Coïsson[1], Kader Telali[1], Benoit Heumez[1], Vincent Lesur[1], Xavier Lalanne[1], and Chang Jiang Xin[2]

[1]Institut de Physique du Globe de Paris, Sorbonne Paris Cité, Université Paris Diderot, CNRS, F-75005 Paris, France
[2]Lanzhou Geomagnetic Observatory, Lanzhou Institute of Seismology, China Earthquake Administration

*Correspondence to:* P. Coïsson (coisson@ipgp.fr)

**Abstract.** During magnetic observatory data acquisition, the data time-stamp is kept synchronized with a precise source of time. This is usually done using a GPS controlled Pulse Per Second (PPS) signal. For some observatories located in remote areas or where internet restrictions are enforced, only the magnetometer data are transmitted, limiting the capabilities of monitoring the acquisition operations. The magnetic observatory in Lanzhou (LZH), China, experienced an interruption of the GPS PPS

in 2013. The data-logger clock drifted slowly in time: in 6 months a lag of 28 s was accumulated. After a reboot on 2 April 2014 the drift became faster, 2 s per day, before the GPS PPS could be restored on 8 July 2014. To estimate the time lags that LZH time-series had accumulated, we compared it with data from other observatories located in East Asia. A synchronization algorithm was developed. Natural sources providing synchronous events could be used as markers to obtain the time lag between the observatories. The analysis of slices of 15 minutes of 1-s data at arbitrary UTC allowed estimating time lags with

an uncertainty smaller than $\sim 10$ s, revealing the correct trends of LZH time drift. A precise estimation of the time lag was obtained by comparing data from co-located instruments controlled by an independent PPS. In this case, it was possible to take advantage of spikes and local noise that constituted precise time-markers. It was therefore possible to determine a correction to apply to LZH time-stamps to correct the data files and produce reliable 1 minute averaged definitive magnetic data.

## 1 Introduction

The Lanzhou Geomagnetic Observatory provides continuous observation of the Earth magnetic field. It is one of the oldest magnetic observatories of China (Yang, 2007) established during the International Geophysical Year initiatives in 1959. It was modernized in 1998, when a collaboration started between the China Earthquake Administration and the Institut de Physique du Globe de Paris (IPGP) (France), which provided new equipment and ensure data processing. Since 2002, this observatory is part of the International Real-time Magnetic Observatory Network (INTERMAGNET) (Love and Chulliat, 2013). Its International

Association of Geomagnetism and Aeronomy (IAGA ) code is LZH and it provides definitive data of 1-minute averages of each magnetic component. From 2009 it produces also 1-second averaged data. The Lanzhou observatory hosts also additional acquisition chains where other magnetometers are usually run in parallel to the main instruments. Absolute measurements are performed by the local staff of the observatory twice per week (Changjiang and Zhang, 2011), while subsequent data



processing and production of Quasi-Definitive (Peltier and Chulliat, 2010) and Definitive (INTERMAGNET, 2012) data is done in France. Due to local regulations, the data are transmitted from the observatory with a delay of one day and operations on the acquisition chain are possible only on-site.

The magnetic instruments include a VM391 3-axis and homocentric fluxgate magnetometer providing 1-second vector data 5 (Chulliat et al., 2009) and a GSM90 scalar magnetometer providing 5-seconds data. Both are controlled by a data logger running on an Acrosser AR-ES0631 fanless embedded system, using a specifically designed software.

## 2 Time-stamp of observatory data

The acquisition system used for recording LZH data from the VM391 and GSM90 magnetometers includes a GPS receiver that provides a Pulse Per Second (PPS) signal for precise time-stamping of the acquired data. Like all recent computers, the 10 data logger is equipped with a material clock: it includes a 64 bits counter that starts when the system is switched on and computes incremental values $C_i$. Its frequency of increment depends on a quartz oscillator that has a nominal frequency of $F_{counter} = 1.19318$ MHz. A virtual clock is also created to provide the UTC time $t_{now}$ when needed. When a time-stamp needs to be generated, the value of the time is obtained from:

$$t_{now} = t_{sync} + \frac{C_{now} - C_{sync}}{F_{counter}} \qquad (1)$$

15 where, $t_{sync}$ is the UTC time provided by the GPS at the emission of its PPS when the datalogger performs its synchronization. At $t_{sync}$ the data logger counter recorded the value $C_{sync}$, and records a value $C_{now}$ at the current epoch.

### 2.1 GPS synchronization and correction of oscillator frequency drift

A GPS antenna is installed on the roof of the observatory, connected to a GPS receiver that provides a PPS signal to the data logger via a RS232 link. The width of the PPS signal can be configured between few $\mu$s to few ms. After every PPS emission, 20 the GPS receiver provides also the complete date in UTC hours through the same link. This time-stamp pertains to its previous PSS and thus it corresponds to an integer number of seconds. This is also the desired time when obtaining magnetometer readings.

Since the frequency of the quartz oscillator depends on its temperature, it is necessary to keep track of the drift of the computed time $t_{now}$ in order to keep the time-stamp of the data logger within an acceptable error (INTERMAGNET, 2012). 25 In order to do that, the data logger regularly acquires a new value $t'_{sync}$ provided by the GPS PPS and compares it with $t_{now}$ computed using Eq. (1):

$$\Delta t = t_{now} - t'_{sync} \qquad (2)$$

This error $\Delta t$ it then used to correct the frequency $F_{counter}$ used to compute $t_{now}$ to maintain $\Delta t = 0$. The value $C_{sync}$ is also updated to the counter value at the time of synchronization. This process is performed 3 times per hour when the PPS signal is available, at minutes 15, 30 and 45, all at 0 s.



**Table 1.** Geographical positions and distances between Lanzhou and the other observatories.

| Observatory | Latitude [°] | Longitude [°] | distance [km] |
|:-----------:|:------------:|:-------------:|:-------------:|
| LZH | 36.087 | 103.845 | - |
| PHU | 21.029 | 105.958 | 1687 |
| DLT | 11.945 | 108.482 | 2724 |
| CYG | 36.370 | 126.854 | 2059 |
| KAK | 36.232 | 140.186 | 3243 |

In case of a failure of the PPS signal, the data logger keeps using the last value of $F_{counter}$ and $C_{sync}$ that were obtained at the last $t_{sync}$. A message is issued in the observatory log file to indicate the failure of the synchronization. These values are kept in the memory of the data logger but are lost when a reboot of the system becomes necessary.

## 2.2 Verification of time-synchronization between different instruments

When we noticed that the time-synchronization using GPS PPS was unavailable for LZH data, we first decided to use data readily available at IPGP or on INTERMAGNET to understand if we could get a reasonable estimate of the time-stamp error of the recorded data. We first selected observatories on the same longitudinal sector as Lanzhou: the nearest observatory available is the one at Phu-Thuy (PHU) in Viet Nam at nearly 1700 km distance. We decided to use a few observatories to inter-compare their time-series, selecting other observatories within 3500 km distance from Lanzhou. We selected also Da Lat (DLT) observatory in Viet Nam, Cheongyang (CYG) observatory in Korea and Kakioka (KAK) observatory in Japan. The details of positions and distances from Lanzhou are shown in table 1. The farthest observatory, KAK, has a longitude distance of $23°$, which corresponds to more than one hour delay in the occurrence of the magnetic field diurnal variation, but it is the only one providing a complete time-series during the whole period of analysis. For the synchronization process we used variational data, i.e. data that were not manually processed to remove spikes and artefacts. In particular, at Lanzhou observatory, quite frequent magnetic perturbations are observed, some due to nearby road traffic and other due to geophysical experiments running on the same site.

Whenever magnetic pulsations were recorded simultaneously at the various observatories, these signals were used to evaluate the time lag between the various time series. Figure 1 shows an example of magnetogram recorded on 6 July 2014, just before the GPS receiver was re-established for LZH data-logger. To compare the data of distant location, each magnetic component time series was first standardized over one day. The top panel of Fig. 1 shows that the diurnal variation exhibits different trends at each observatory, since the Solar quiet (Sq) currents characteristics depend on the magnetic latitude of the observatories. On that day, around 11 UTC, a fast increase of the X component appears simultaneously in all observatories, lasting about 20 minutes. This synchronous event is seen earlier in LZH time-series, indicating that the data-logger clock was running faster than in the other sites. This kind of ramp is however not usable to estimate a time lag with a needed precision of the order





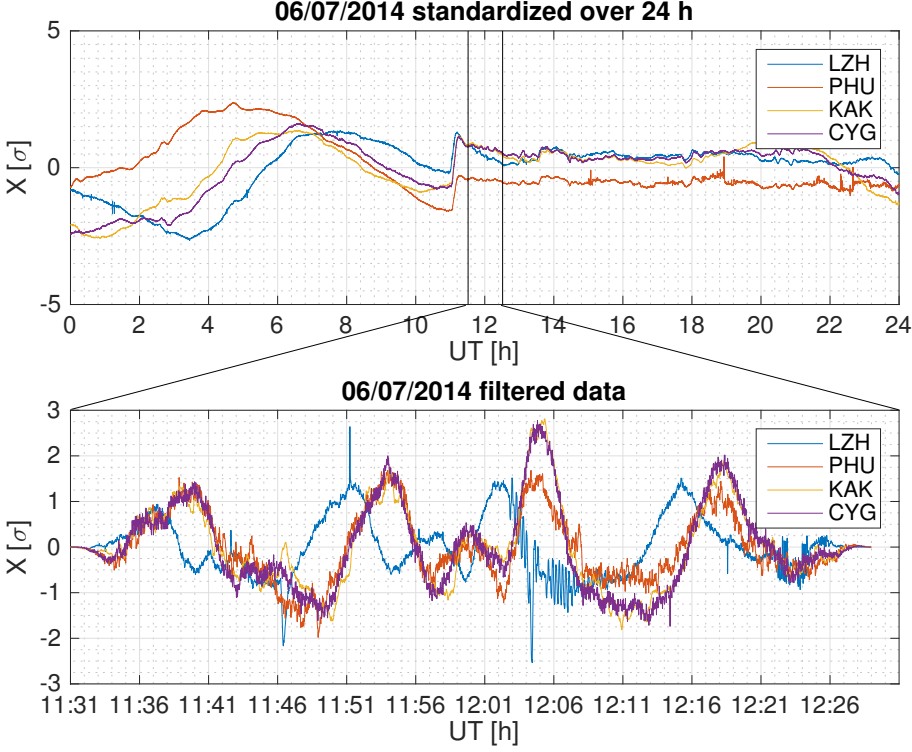

**Figure 1.** Time-series of the normalized magnetic X component of LZH, PHU, KAK, CYG observatories over 24 hours (top) and a zoom over one hour between 11:30-12:30 UTC, when magnetospheric activity is observed simultaneously at all observatories. The uncorrected data of LZH appear to anticipate this activity by about three minutes.

of 1 s: it is too long and it has a total duration that is not exactly the same at all sites. This event was followed by magnetic pulsations producing numerous oscillations with periods of a few minutes, clearly seen at all sites.

The bottom panel of Fig. 1 shows the data recorded between 11:30 UTC and 12:30 UTC. These time series have been pro-
5 cessed to allow further analysis: first each time series in this window has been detrended and standardized. Then, a polynomial of order 4 has been fitted to the standardized data and removed. Finally, a Tukey windows has been applied, to force the edges of the time series to be close to 0. In this figure, a very similar pattern of wave activity is seen at all observatories and a good synchronization is obtained in all distant sites. It can be clearly observed that LZH time series was incorrectly labelled and preceding the others by nearly three minutes.

10 To obtain the estimation of the time lag, the filtered time series of each measured component at all observatories have been cross-correlated in pairs. An example for that same day, during local night, is shown in Fig. 2. The cross correlation with the second acquisition system available at Lanzhou is also shown in this figure. The cross-correlation curves show that it is easier to estimate the lags between time series using the X and Z components, because they present quite sharp peaks. Observatories




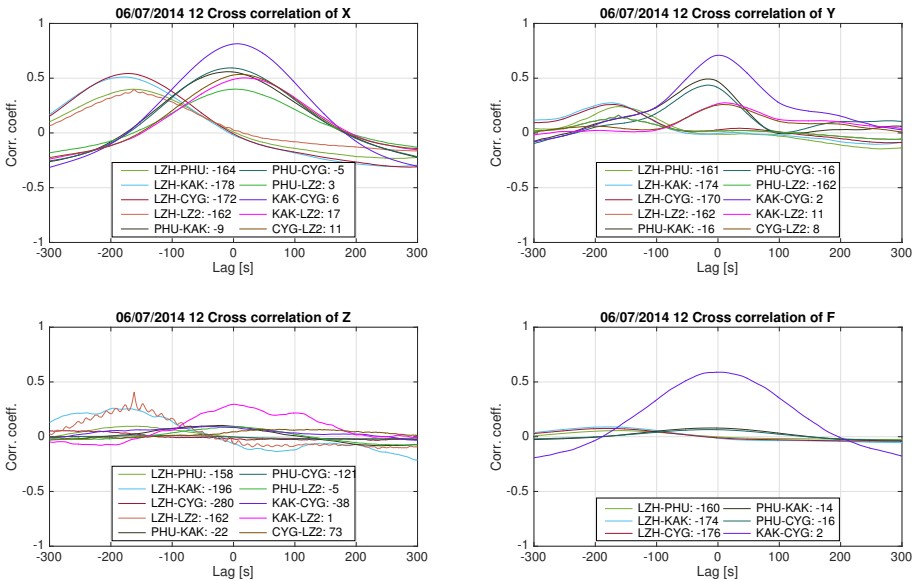

**Figure 2.** Cross correlation between all analysed observatories over one hour of data (11:30-12:30 UTC) on 6 July 2014, one of the last days without GPS synchronisation for LZH data logger. The second acquisition system of Lanzhou observatory is labelled LZ2.

with reliable time-stamp show a cross-correlation lag near to 0 s, but it can sometimes exceed 10 s, if the cross correlation peak is wide. Data from all observatories correlated with LZH time-series agree to estimate the lag between 162 and 196 s.

From this first analysis, it appeared clearly possible to use distant observatories for verifying the data synchronization, but
the precision is not sufficient precision for the purpose of correcting the data time-stamps. We computed an estimate of the time lag for each day between March 2013 and July 2014, obtaining coherent trends for all pairs of observatories. Fig. 3 shows the time lags obtained using Kakioka observatory data for each measured component. The time lag estimation is very noisy on Z and Y components and the lower sampling of F at Lanzhou produces a curve that is more spread than for the X component. Nevertheless the variation of the time lag during the year is evident. Changing UTC of analysis or increasing the length of the
cross-correlation time window affect strongly the spreading of the lags. The night-time hours are the ones where the lags are obtained with lower noise. A full set of figures for lags computed at each UTC is provided as supplementary material to the article. The analysis of the evolution of the lags through the whole period reveals that two different trends are observed: a slow variation during 2013 and up to April 2014 and fast variations from April to July 2014, after the data-logger was rebooted and the correction for the oscillator frequency was lost.
It was therefore decided to use the data of the second acquisition chain available in Lanzhou for computing the time lags suitable for correcting the time-stamps. Two different instruments were available on the second acquisition chain, one in 2013 and another in 2014, allowing to generate a complete data-set for comparison. The resulting curves, shown in Fig. 4 present coherent lags values for all the three components X, Y, Z, and the F component is more noisy due to the lower sampling rate of the scalar magnetometer. Particularly the Z component exhibits a very small dispersion of data and just few outliers: it is



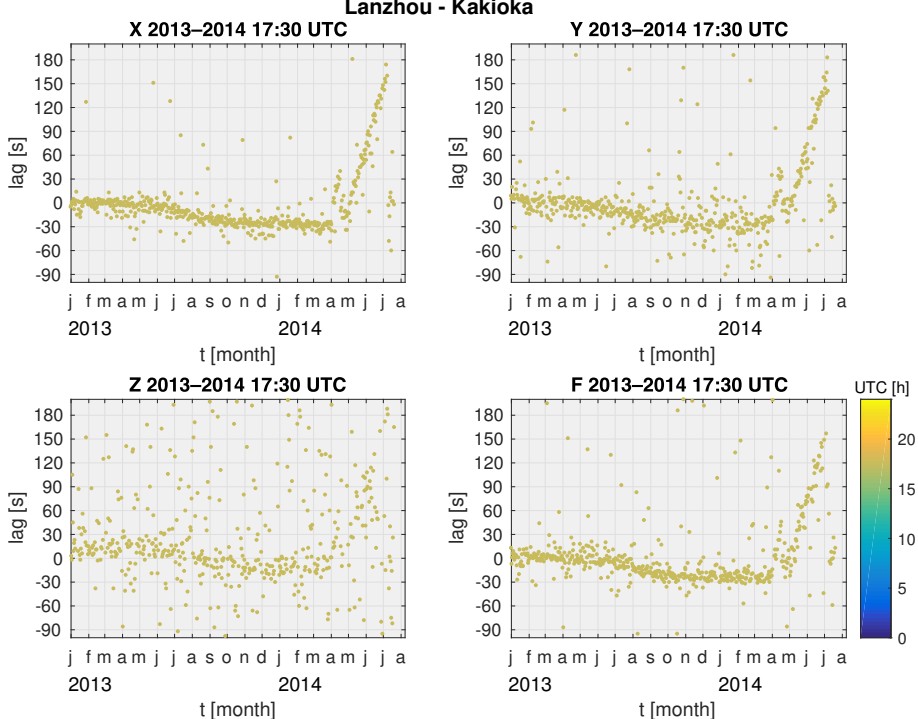

**Figure 3.** Time lags calculated every day between 17:22 and 17:37 UTC comparing Lanzhou and Kakioka data for each component of the magnetic field X, Y, Z and F. F is measured every 5 s at Lanzhou.

the component where the amplitude of the spikes occurring in this observatory is larger. These spikes, lasting 3 to 5 s, are local phenomena that can be used to precisely estimate the lag between the two time series since they produce a very narrow peak in the cross correlation curve (see Fig. 2).

5 ### 2.2.1 Correction of data time-stamp

After computing all the time lags, it was decided that only the period up to 2 April 2014 was suitable for time-stamp correction, since the clock drift was very slow during that time. Data for the period between 2 April and 8 July 2014, when the time drift was of 2 s per day, will not be published as definitive data. A new set of corrected LZH 1 s data files was then generated using the computed time lags. A single daily correction value was used since the drift of the clock during one day was always smaller

10 than 0.5 s and the purpose of the correction was to produce corrected 1-minute data files. The corrected 1 s data files were averaged to compute 1 minute data files following the usual INTERMAGNET recommendations (INTERMAGNET, 2012). The following baseline processing allowed to generate corrected 1 minute definitive data for LZH observatory.


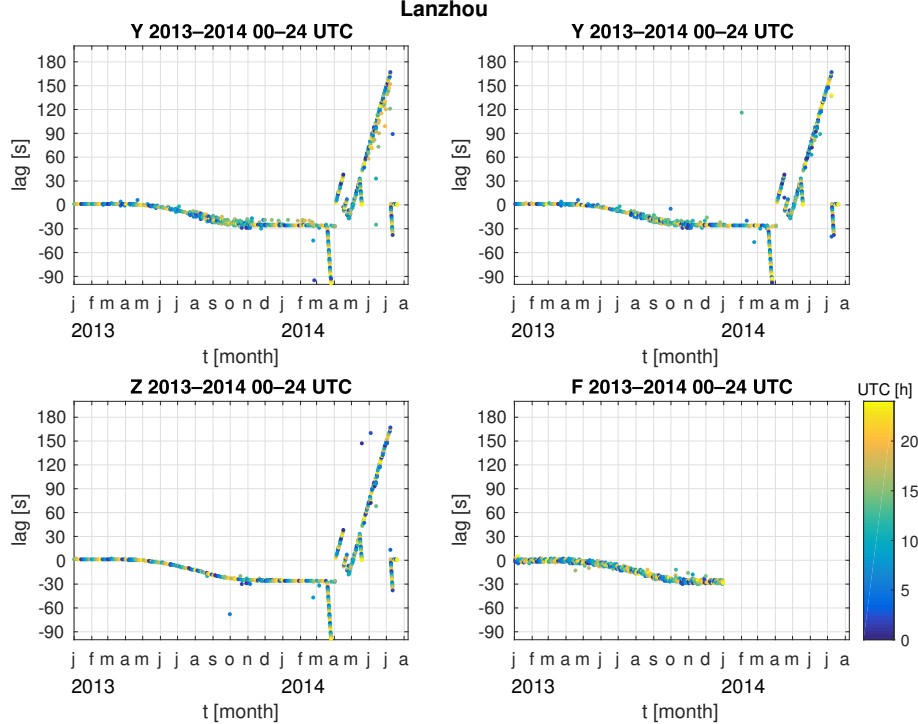

**Figure 4.** Time lags calculated for every hour using the data of the two acquisition chains available at Lanzhou observatory for each component of the magnetic field X, Y, Z and F. F is measured every 5 s at Lanzhou, the second scalar magnetometer was not available in 2014.

## 3 Discussion

The time-synchronization of LZH VM391 and GSM90 instruments was lost on 7 March 2013, but it was noticed only one year later since the drift of the data logger clock was very slow. The acquired data had accumulated a lag of -28 s compared to UTC time. This drift is relatively low, thank to the correction of the oscillator frequency that is done by the data-logger. During the first two months, only -2 s of lag were accumulated. In the period between June and September the negative lag increased at a faster rate and reached -24 s. Afterwards again the negative lag increased more slowly, to reach a stable level of -27 s in December 2013 and was kept at this level up to the beginning of April 2014, when the data logger was rebooted.

The most significant part of the lag was accumulated during the summer months, when the temperature of the data logger was the highest (Fig. 5). Though a correlation between the quartz frequency and the temperature of the data logger is expected, it appears to be non-linear. The temperature of the data logger follows a smooth seasonal trend, with a lag of a couple of months with respect to the ambient temperature. Only when the temperature was exceeding more than 5°C the one at the time when $F_{counter}$ was last estimated, the clock drifted at a higher pace. When the temperature returned again below this 5°C difference, the clock nearly stopped drifting. After the reboot of the data logger on 2 April 2014, the clock correction was




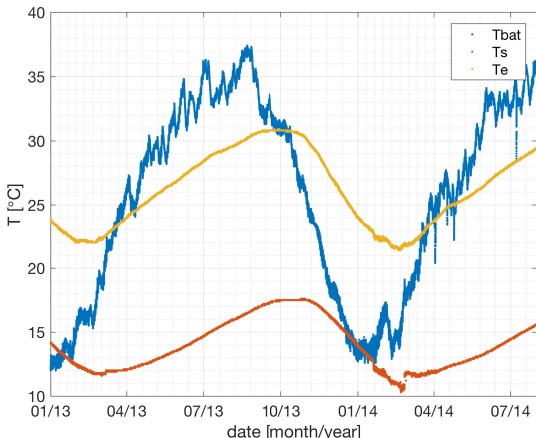

**Figure 5.** Seasonal variation of the battery temperature (Tbat), magnetometer sensor temperature (Ts) and data logger temperature (Te) of LZH acquisition chain during 2013 and 2014. The battery temperature is measured every minute, while the other temperatures are measured every 10 minutes.

erased and no longer available in the data logger memory. It was not possible for the data logger to compute a new correction, since the PPS was not available due to the GPS failure. The data logger clock started drifting with a constant rate of about 2 s per day. Additional reboots were done afterwards, and at every reboot the clock counter started at a different value producing the saw-tooth behaviour that can be observed in the lags shown in Fig. 4. To avoid having similar issues in the future, some options are possible:

- use a data logger with a Temperature Compensated Crystal Oscillator (TCXO),

- improve the software for clock correction to save the correction to the quartz frequency so that they are available even after a reboot of the data logger and possibly include also temperature-correction,

- improve the monitoring tools of the observatory so that similar failures could be detected more easily. It should be pointed out that this particular situation occurred because, at that time, the log files were not routinely transmitted along with the data files.

LZH monitoring data are now regularly transmitted to IPGP to prevent future occurrences of a similar situation.

## 4 Conclusions

The GPS time-synchronization of LZH magnetic observatory was lost on 7 March 2013. Over one year, the time-stamp attribute to the acquired data had accumulated a lag of -28 s compared to UTC time. This drift is low, thank to the correction of the oscillator frequency that is done by the data-logger. It is possible to confidently correct the time-stamps of the 1-second





acquisitions to produce 1-minute definitive data, the official INTERMAGNET products. It has been proven that comparing the time series of one observatory at mid-latitude with the time series of observatories within 3500 km range, it is possible to detect the correct trend of time drift. This comparison is more effective during hours when there is low diurnal variation and is better

identified on the magnetic X component. This is an effective way to verify the stability of clocks in un-manned acquisition systems that cannot be monitored in real time. To be able to construct a precise time-correction function it is preferable to use another acquisition system located in the same premises, like in Lanzhou observatory. In this case, all spikes caused by local activity, that are usually removed from magnetic definitive data, provide short signals that facilitate obtaining a precise time-correction. In the case of Lanzhou, the vertical component of the magnetic field is the most affected by spikes and could

be used to correct the synchronization of the time-series. The time-stamps for the whole period between March 2013 and 8 April 2014 were corrected and average 1 minute values definitive data could be produced.

## 5   Data availability

Magnetic observatory data used for this paper are available at the Bureau Central du Magnetisme Terreste server (www.bcmt.fr) and on INTERMAGNET server (www.intermagnet.org).

*Author contributions.*   P. Coïsson analysed the data, produced the corrected files and wrote the article, B. Heumez identified the problem and produced definitive data, K. Telali and X. Lalanne developed LZH observatory acquisition system and interpreted the instrument response, V. Lesur validated the methodology, C.J. Xin processed the data. P. Coïsson prepared the manuscript with contributions from all co-authors.

*Competing interests.*   The authors declare that they have no conflict of interest.

*Acknowledgements.*   The results presented in this paper rely on the data collected at Lanzhou, Phu Thuy, Da Lat, Kakioka and Cheongyang

observatories. We thank IPGP, Chinese Earthquake Administration, Viet Nam Academy of Science and Technology, Japan Meteorological Agency, Korea Meteorological Administration for supporting its operation and INTERMAGNET for promoting high standards of magnetic observatory practice (www.intermagnet.org). This is IPGP contribution XXXX.





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
