# Peer review of "Time-stamp correction of magnetic observatory data acquired during unavailability of time-synchronization services"

_Geoscientific Instrumentation, Methods and Data Systems, 2017_

## Referee Comment (RC1) · Anonymous Referee #1 · 29 Mar 2017

GENERAL COMMENTS:

The subject presented in this paper could be very informative and interesting for people who conduct observations in geomagnetic observatories. In this study the authors propose a new scheme for time-stamp correction of magnetic observatory data during unavailability of time-synchronization services. They showed that a precise time-stamp correction of the main variometer system can be obtained by comparison with data from the supplement system with accurate time-stamps. The authors clearly demonstrated how to obtain a reliable time-stamp correction by taking advantage of spikes and local noise, and applying the cross correlation analysis on the time series from co-located instruments. Thanks to the clear description and computational simplicity, the method

can be easily applied in other word-wide observatories that have experienced similar problems.

Further, it is showed and illustrated that it is possible to detect a time lag in LZH data through comparison with the data from surrounding observatories. However, in LZH case the separation from nearby observatories is quite big and it is not possible to obtain the time-stamp correction with sufficient precision, i.e. scattering of the time lags in Fig. 3 (and supplement figures) is too big. Nevertheless, maybe their method could be successfully applied in a case when we have smaller separations between observatories (e.q. Europe). This could be particularly useful in cases where observatories do not have two acquisition systems controlled by independent PPS.

The article is well written, understandable and appropriately referenced. In my opinion the article is suitable for publication in GI and I recommend that the article can be published after minor corrections and clarifications of doubts.

SPECIFIC COMMENTS:

1) In Fig. 1 and 2, (06.07.2014) we have negative values of the time lags, i.e. LZH time-series precede the others by nearly three minutes. This means the time lag has negative sign and this is correctly labelled on legends in Fig 2. On the other hand in Figs. 3 and 4 in the same period, July 2014, the time lag has positive sign. You should check sings on these figures. According to Fig. 1 and 2 I would expect reverse signs on y-axis in Figs. 3 and 4. If you change signs on Figs. 3 and 4 pay attention to do this also in Discussion and Conclusion section.

2) To clearly show dependence between the clock drift and the temperature difference (between the ambient and data logger temperature) I recommend changes on Fig. 5. Instead the sensor temperature (which is irrelevant for discussion) to plot the difference between the battery temperature and data logger temperature. The y scaling for this curve can be placed on the right side of the plot.

3) Did you note any problems in absolute measurements, i.e. with base values during the drift period? At least for the period Apr-Aug 2014 when time lags were higher I would expect more scattered observations, especially in the H component. I presume if you use variometer data with a few minutes time lag, to derive base values, this could introduce a few nT errors. Of course this also depends on the local geomagnetic activity during observational times, but in general I would expect systematic increase in scattering of the base values parallel with an increase of a time lag in recordings. If this was the case, maybe this fact should be mentioned in the text.

TECHNICAL CORRECTIONS:

Page 1, line 18: ensure -> ensures, is part -> is a part, line 20: (IAGA ) -> (IAGA)

Page 4, line 6: Tukey windows -> Tukey window

Page 5, Line 5: "...the precision is not sufficient precision for the purpose..." -> "...the precision is not sufficient for the purpose ...", line 19: few -> a few

Page 7, Figure 4: subplot left-top: "Y 2013..." -> "X 2013...", Lines 12-13: "Only when the temperature was exceeding more than 5°C the one at the time when F_counter was last estimated, the clock drifted at a higher pace." -> "Only when the temperature difference was exceeding more than 5°C at the time when F_counter was last estimated, the clock drifted at a higher pace."

Page 9, line 9: the most affected -> most affected

Mostly you use "data logger", for consistency you should correct:

Abstract, line 5 "data-logger", page 2, line15 "datalogger", also page 3, line 20, line 24, page 5, line 14, page 7, line 5.

Also use "cross correlation" or "cross-correlation" everywhere in the text.

Page 7, line 7: -27 -> -28 (everywhere else in the paper you are talking about lagging of 28 s)

In my opinion the term "acquisition chain" could be replaced with "acquisition system".

---

## Referee Comment (RC2) · Anonymous Referee #2 · 17 Apr 2017

GENERAL COMMENTS:

This study is very useful and interesting for geomagnetic observatories and data users. The authors propose the method which correct time-stamp using time-series of other observatories or the second acquisition system with GPS synchronization. This might be good method to ensure or correct the time-stamp of data from observatories with un-manned acquisition system or those without the second acquisition system.

However, I think that quantitative discussion about accuracy and precision of time correction value is insufficient. It is necessary to show accuracy and precision of the time correction value using time-series which have GPS synchronization at both of the LZH and the reference stations. The accuracy and precision may depend on position of

reference observatories or time of analysis.

I would recommend the article for acceptance after dealing with the issue of accuracy and precision.

SPECIFIC COMMENTS:

1) I have a concern about cross-correlation using local signals. In my understanding, when the time series at LZH and LZ2 are cross-correlated, signals from road traffic are also computed. I guess that there might be a time lag if the sensors of LZH and LZ2 is not on the line perpendicular to the road.

For example:

- Sensor of LZH is 50 m away from that of LZ2 to North.

- There is a road going north and south.

- A car go to north with 10 m/s.

In above case, there will be the computed time lag of 5 s, even though LZH and LZ2 have GPS synchronization.

In addition, what does the oscillation of Z component mean in Figure 2? Do each narrow peaks represent car signals?

2) It is better to write magnetic coordinates of each observatory in Table 1, since the Sq currents are discussed in line 22 of page 3.

3) Please include enough information about making "A single daily correction value" in the section 2.2.1. Which data did you use, LZ2 or KAK? In the case of LZ2, there are 24 time lags per day. In the case of KAK, there is one time lag per day which have large dispersion. How did you calculate "A single daily correction value"?

4) Please describe the required time accuracy for making 1-minute definitive data, citing the INTERMAGNET Technical Reference Manual.

TECHNICAL CORRECTIONS:

Page 2, line 21: PSS -> PPS

Page 3, lines 12, 13: Is "longitude distance" a difference between longitude of LZH and that of other observatory? According to Table 1, the longitude distance of KAK is 36 degrees and the time difference of KAK is two hours.

Page 4, Figure 2: To make it easier to discriminate the different lines in Figure 2, I recommend that you use some type of lines, e.g. dashed lines. It is difficult for me to distinguish some lines in Figure 2.

Page 8, Figure 5: To make it easier to recognize these lines in Figure 5, I recommend that you change the markers of legend bigger or longer.

---

## Author Comment (AC1) · 9 Jun 2017

Thank you for the positive comments to our work, all questions contributed to greatly improve the article. We provide here the answers to the points raised.

**Comment:** *1) In Fig. 1 and 2, (06.07.2014) we have negative values of the time lags, i.e. LZH time-series precede the others by nearly three minutes. This means the time lag has negative sign and this is correctly labelled on legends in Fig 2. On the other hand in Figs. 3 and 4 in the same period, July 2014, the time lag has positive sign. You should check sings on these figures. According to Fig. 1 and 2 I would expect reverse*

*signs on y-axis in Figs. 3 and 4. If you change signs on Figs. 3 and 4 pay attention to do this also in Discussion and Conclusion section.*

**Answer:** Indeed in the original manuscript there was an ambiguity on the definition of lag. Figure 2 was showing the lag obtained from cross correlation, while Figures 3 and 4 showed lags as estimated time-correction for LZH time-stamp. Both had been labelled "lag", but they have opposite sign.

In the revised version of the article the lag has been defined explicitly lag=$t_{\mathrm{LZH}} - t_{\mathrm{r}}$ and to avoid confusion, all figures have been redrawn to show lags following this definition. The whole text have been checked for consistency.

**Comment:** *2) To clearly show dependence between the clock drift and the temperature difference (between the ambient and data logger temperature) I recommend changes on Fig. 5. Instead the sensor temperature (which is irrelevant for discussion) to plot the difference between the battery temperature and data logger temperature. The y scaling for this curve can be placed on the right side of the plot.*

**Answer:** Thank you for this suggestion, the figure has been redrawn keeping only the most relevant quantity (energy card temperature). We do not have a record of the ambient temperature around the data logger, therefore we can only analyse the energy card temperature (that was indicated as battery temperature in the original submission). What was indicated as data logger temperature is the temperature of the electronic components a few meters away from the sensor, but in another building.

In the new figure we plotted only the energy card temperature, highliting its value at the time of loss GPS failure with an horizontal line. The caption and the text of the article were all changed accordingly to explain more clearly which temperature is shown..

**Comment:** *3) Did you note any problems in absolute measurements, i.e. with base*

*values during the drift period? At least for the period Apr-Aug 2014 when time lags were higher I would expect more scattered observations, especially in the H component. I presume if you use variometer data with a few minutes time lag, to derive base values, this could introduce a few nT errors. Of course this also depends on the local geomagnetic activity during observational times, but in general I would expect systematic increase in scattering of the base values parallel with an increase of a time lag in recordings. If this was the case, maybe this fact should be mentioned in the text.*

**Answer:** During the first period, from March 2013 to February 2014, the time lag on the data time-stamp did not produce a significant scattering of the baseline. In LZH, for baseline measurements each measure is taken at second 0 of each minute, using a clock available in the absolute measurements room. At every measurement session 10 measurements of the magnetic field intensity $F$ are also recorded in the absolute measurements room, to compute $\Delta F$ with the sensors room. During the whole period under analysis, the $\Delta F$ remained quite stable around $1.5$ nT, even during the period when the clock was drifting faster.

For the period between April and July 2014, since the acquisition system clock drift was too fast to provide a good correction of the time-stamp, we decided not to produce definitive data. Nevertheless absolute measurements were done routinely and the baseline was computed. It appears that there is a larger spreading of the measurements during this period (about 3.5 nT peak to peak on H component and 2 nT on Z component). This larger spreading starts in March 2014, before the reboot of the data logger and does not disappear at the time when the GPS PPS was re-established, lasting till September 2014. Thus it is not clear if it is correlated with the time-stamp or with the seasonal variation of the temperature in the absolute measurements room.

We do not think that this specific point should be discussed in the article, since there is not a well-defined correlation between the time-stamp evolution and the spreading of absolute measurements.

**Comment:** *Page 1, line 18: ensure → ensures, is part → is a part, line 20: (IAGA ) → (IAGA)*

**Answer:** These mistakes were corrected according to the reviewer suggestions.

**Comment:** *Page 4, line 6: Tukey windows → Tukey window*

**Answer:** This typo has been corrected.

**Comment:** *Page 5, Line 5: ". . .the precision is not sufficient precision for the purpose. . ." → ". . .the precision is not sufficient for the purpose . . .", line 19: few → a few*

**Answer:** These errors have been corrected.

**Comment:** *Page 7, Figure 4: subplot left-top: "Y 2013. . ." → "X 2013. . ."*
   **Answer:** The figure has been corrected.

**Comment:** *Lines 12-13: "Only when the temperature was exceeding more than 5°C the one at the time when F counter was last estimated, the clock drifted at a higher pace." → "Only when the temperature difference was exceeding more than 5°C at the time when F counter was last estimated, the clock drifted at a higher pace."*

**Answer:** The long and difficult sentence was broken into two separate shorter sentences: "It appears that the difference between this temperature and its value at the time when $F_{counter}$ was last estimated played an important role in the clock drift. Only when this temperature difference was exceeding more than $5°C$, the clock drifted at a higher pace."

**Comment:** *Page 9, line 9: the most affected → most affected*

**Answer:** The sentence was modified into "is the one most affected".

**Comment:** *Mostly you use "data logger", for consistency you should correct: Abstract, line 5 "data-logger", page 2, line15 "datalogger", also page 3, line 20, line 24, page 5, line 14, page 7, line 5.*

**Answer:** The text was revised using "data logger" everywhere.

**Comment:** *Also use "cross correlation" or "cross-correlation" everywhere in the text.*

**Answer:** The text was revised using "cross-correlation" everywhere.

**Comment:** *Page 7, line 7: -27 → -28 (everywhere else in the paper you are talking about lagging of 28 s)*

**Answer:** Actually in December the delay was 27 s. It remained stable all through the early months of 2014 and it is only during March 2014 that the last second of delay was accumulated. This value was not changed, but the text was modified to explain this point:

- Abstract: the value was changed into 27.

- Discussion: the sentence "and was kept at this level up to the beginning of April 2014" was changed into "It remained at this level up to March 2014, when a slowly increase started again. 28 s lag was reached before the data logger reboot on 2 April 2014."

**Comment:** *In my opinion the term "acquisition chain" could be replaced with "acquisition system".*

**Answer:** Following the reviewer suggestion, "acquisition chain" was replaced with "acquisition system" throughout the text.

Interactive
comment

---

## Author Comment (AC2) · 9 Jun 2017

Thank you for evaluating positively this work. The comments and remarks you raised were all relevant and we followed them to improve the manuscript and correct it.

**Comment:** *This study is very useful and interesting for geomagnetic observatories and data users. The authors propose the method which correct time-stamp using time-series of other observatories or the second acquisition system with GPS synchronization. This might be good method to ensure or correct the time-stamp of data from observatories with un-manned acquisition system or those without the second acquisi-*

[Figure]

*tion system.*
*However, I think that quantitative discussion about accuracy and precision of time correction value is insufficient. It is necessary to show accuracy and precision of the time correction value using time-series which have GPS synchronization at both of the LZH and the reference stations. The accuracy and precision may depend on position of reference observatories or time of analysis. I would recommend the article for acceptance after dealing with the issue of accuracy and precision.*

**Answer:** We agree with the reviewer comment, that it is necessary to control the effectiveness of the method during times when both compared datasets have a correct time-stamp. This was done in this work for the period between 1 January and 7 March 2013, before the interruption of the GPS PPS. The results can be seen in Figures 3 and 4, that start on 1 January. In the case of the co-located instruments, the computed time lags in this period are almost always 0 s and in rare cases 1 s (average 0.1 s, standard deviation 0.3 s on X Y and Z component). In the case of the comparison with Kakioka, the spread of data is larger and it depends on the local time at the observatories, the best results are obtained on X component at 5:30 UTC when an average of -3 s and standard deviation of 11 s. A table containing the whole statistics has been added in the supplementary information.

Moreover, Figure 2 shows all the cross-correlations between each pair of observatories, when only LZH was having a failing GPS synchronization. This figures shows that the cross-correlation peak is large, centred near 0, but the estimated lags can be larger than 10 s, especially for magnetic component that are not presenting a sharp peak in the cross-correlation (Y, Z, F).

Since we understand that it is not explicit in the text of the article, we included in the abstract the date when the GPS PPS was interrupted, 7 March 2013 and inserted in section 2.2 that the period over which we computed the time lag started on 1 January 2013 and we added the results of the statistics: "During the period between 1 January 2013 and 7 March 2013, when the GPS synchronization of LZH observatory was still

operational, the cross-correlation of one hour X values around 5:30 UTC resulted in an average time lag of -3 s with a standard deviation of 11 s."

SPECIFIC COMMENTS:

**Comment:** *1) I have a concern about cross-correlation using local signals. In my understanding, when the time series at LZH and LZ2 are cross-correlated, signals from road traffic are also computed. I guess that there might be a time lag if the sensors of LZH and LZ2 is not on the line perpendicular to the road. For example:*

- *Sensor of LZH is 50 m away from that of LZ2 to North.*

- *There is a road going north and south.*

- *A car go to north with 10 m/s. In above case, there will be the computed time lag of 5 s, even though LZH and LZ2 have GPS synchronization. In addition, what does the oscillation of Z component mean in Figure 2? Do each narrow peaks represent car signals?*

**Answer:** This comment is important and indeed signals from moving trucks are delayed according to the relative geometry of the road and sensors. If the only source of local noise is road trafic, the delay of detection in each site has to be taken into account. In the specific case of LZH observatory, the two instruments are located within the same room, at a distance of about 3 m. Near LZH sensor building there is a small agricultural road whose traffic is only one among various sources of spikes. Most of recorded spikes last just a couple of seconds, as it can be seen in Figure 1 and they are the ones producing the narrow peak on Z component. Other instruments in the observatory produce a different kind of signal: many times per day regular oscillations that last few minutes are generated, also seen in Figure 1. The short spikes are usually more pronounced on the Z

component, thus the cross-correlation present a narrow peak, while the oscillating signals produce oscillations also in the cross-correlations function with peaks corresponding to multiples of the period of these signals. We also found in LZH data perturbations lasting up to about 1 minute, that could be related to moving vehicles, but we could not verify it. We are anyway confident that the issue of a moving source is not affecting this particular analysis.

To better explain LZH perturbations, the sentence "In particular, at Lanzhou observatory, quite frequent magnetic perturbations are observed, some due to nearby road traffic and other due to geophysical experiments running on the same site." has been changed: "at Lanzhou observatory, quite frequent magnetic perturbations are observed of various durations, from a few seconds up to a couple of minutes. The longest are due to nearby road traffic and to geophysical experiments running on the same site."

It has also been specified "the second acquisition system available in Lanzhou inside the same room".

**Comment:** *2) It is better to write magnetic coordinates of each observatory in Table 1, since the Sq currents are discussed in line 22 of page 3.*

**Answer:** The table has been modified to include both geomagnetic and geographic coordinates and the distance with respect to LZH.

**Comment:** *3) Please include enough information about making "A single daily correction value" in the section 2.2.1. Which data did you use, LZ2 or KAK? In the case of LZ2, there are 24 time lags per day. In the case of KAK, there is one time lag per day which have large dispersion. How did you calculate "A single daily correction value"?*

**Answer:** We used the data from the LZ2 instrument to compute the time-correction. The clock drift was sufficiently slow to be largely below 0.5 s during a single day, apart during the period April-July 2014 for which we decided not to produce definitive data. During the month of faster clock drift, August 2013 1 s was accumulated in 4 to 5 days for a total of 7 s in a month. As the lags computed on Z component present a very smooth curve, it was possible to fit an hyperbolic tangent that follow closely the computed cross-correlation lags. The value of that curve at 12 UTC was used for the time-stamp correction. We recall that the aim of this correction is to provide 1-minute averages definitive data.

Some additional details have been added in the text : "A single daily correction value was used, based on the second instrument available in Lanzhou observatory. This correction value was calculated for 12 UT, following a smooth hyperbolic tangent fitted to the calculated hourly delays."

**Comment:** *4) Please describe the required time accuracy for making 1-minute definitive data, citing the INTERMAGNET Technical Reference Manual.*

**Answer:** INTERMAGNET Technical Reference Manual indicates that the data logger clock should have a drift below 5 seconds per month. With the time correction we applied we are well below this limit during the whole period when we applied the correction.

To avoid confusion, we followed the suggestion and indicated explicitly this value in the text : "This choice was possible since the drift of the clock was always well below the 5 seconds/month recommended by INTERMAGNET for computing 1 minute values. The corrected 1 s data files were averaged to compute 1 minute data files following the INTERMAGNET recommendations for data filtering."

TECHNICAL CORRECTIONS:

**Comment:** *Page 2, line 21: PSS → PPS*

**Answer:** The typo was corrected.

**Comment:** *Page 3, lines 12, 13: Is "longitude distance" a difference between longitude of LZH and that of other observatory? According to Table 1, the longitude distance of KAK is 36 degrees and the time difference of KAK is two hours.*

**Answer:** Thank you for pointing out this mistake: as the reviewer correctly pointed out, there are more than 2 hours local time difference between KAK and LZH. The text has been corrected accordingly.

**Comment:** *Page 4, Figure 2: To make it easier to discriminate the different lines in Figure 2, I recommend that you use some type of lines, e.g. dashed lines. It is difficult for me to distinguish some lines in Figure 2.*

**Answer:** We regret that this figure contains too many lines to provide an easy readability of all lines. We tried plotting some as dashed lines, but the result was not satisfactory. The value of the lag (position of the maximum of the cross-correlation) is indicated in the legend and the reason of including this figure was to show that there can be a large uncertainty on the estimate of the lag because of the width of the cross-correlation peak.
We did not modify this figure.

**Comment:** *Page 8, Figure 5: To make it easier to recognize these lines in Figure 5, I recommend that you change the markers of legend bigger or longer.*

**Answer:** Sorry for this inconvenience, the small dots were the default symbol appearing in figures generated without connection line between points. We modified the way the figure was realized (also following the suggestions of the other reviewer) and the new figure is easier to interpret, we kept only the temperature of the energy card (the other temperatures are from components in another building) and we included a line showing the temperature at the time of GPS failure.